# The Material Heterogeneity Effect on the Local Resistance of Pultruded GFRP Columns

**DOI:** 10.3390/ma17010153

**Published:** 2023-12-27

**Authors:** Yongcheng Zhu, Viktor Gribniak, Chaofeng Ding, Hua Zhu, Baiqi Chen

**Affiliations:** 1Department of Steel and Composite Structures, Vilnius Gediminas Technical University (VILNIUS TECH), Sauletekio Av. 11, LT-10223 Vilnius, Lithuania; yongcheng.zhu@outlook.com; 2Department of Civil Engineering, Yancheng Institute of Technology, 1st Xiwangdadao Road, Yancheng 224051, China; yczh716@ycit.cn (H.Z.); baiqi.chen@outlook.com (B.C.); 3Changzhou Institute of Building Science Group Co., Ltd., 288 Changjiang Middle Road, Changzhou 213002, China; dingchaofeng@czjky.wecom.work

**Keywords:** pultruded profile, compression test, short column, local failure, support joint, numerical modeling

## Abstract

Pultruded GFRP (glass fiber-reinforced polymer) materials are widely used in structural engineering because of their lightweight, corrosion immunity, and electromagnetic transparency. However, the design of load-bearing components facing substantial compressive stresses, e.g., columns, must be more stringent than steel structures due to excessive deformability, material heterogeneity, and vulnerability to stress concentration. This manuscript investigates the failure performance of locally produced GFRP materials, focusing on the material heterogeneity effect on the mechanical resistance of a support joint of a pultruded tubular GFRP column. This experimental campaign employs relatively short rectangular profile fragments to isolate the support behavior and verify a simplified numerical finite element model, which neglects the nonlinearity of GFRP material. This work determines the material failure mechanisms behind the mechanical performance of pultruded profiles subjected to longitudinal compression for various column lengths.

## 1. Introduction

Pultruded GFRP (glass fiber-reinforced polymer) materials are widely used in construction due to their lightweight, electromagnetic transparency, high strength, and simple manufacturing [1]. Especially in bridges, building reinforcement, and marine construction, pultruded GFRP has emerged as the optimal alternative to steel due to its excellent mechanical performance and environmental resistance [2]. These materials are typically employed as reinforcement (bars, sheets, and laminates) and pultruded load-bearing components (profiles) in combination with concrete [3,4,5,6]. The market share of FRP (fiber-reinforced polymer) profiles has increased rapidly in the last decade to reach USD 15.3 billion, which is 6.4% of the construction market [7]. Vedernikov et al. [2] stated that traditional construction materials such as concrete, steel, and wood are gradually losing positions in many markets and are being replaced with high-performance FRP composites.

Still, the high deformability, failure brittleness, and creep rupture turn pultruded GFRP components into stability losses, significantly affecting their load-bearing capacity, especially in long-term conditions [8,9]. Due to the significant elongation to failure permitted by the fibers and resin, the composite material retains linear elasticity even after substantial deformations. This limitation sets GFRP composites apart from conventional materials like steel and concrete, which typically yield or crack before failure [10,11]. This feature also leads to pultruded GFRP profiles failing to work with concrete as composite structures in practical engineering applications [12,13,14]. Thus, compared to steel, stricter design considerations for the cross-sectional dimensions of GFRP components are required [15,16].

The pultrusion process can be simplified into three steps: first, impregnating fibers in resin material; second, heat curing the mixture using a mold; and finally, cutting the manufactured profile into the desired length components. However, simple pultrusion produces 1D (oriented in the pultrusion direction) filaments’ structure, which does not resist transverse stresses [17]. Due to the differences in the mechanical performance of the fibers (such as glass or carbon filaments) and matrix (primarily epoxy resin), heterogeneity is a fundamental property of pultruded composite materials. It suggests that the internal fiber distribution within pultruded FRP materials exhibits a certain level of randomness. Furthermore, this weakness causes the pultruded FRP profile to separate into a set of plates prone to buckling, which was never considered in steel structures. However, the heterogeneity and lack of transverse stress confinement of pultruded profiles are typically neglected in engineering applications, considering FRP is an anisotropic and perfectly elastic material [4,18]. Therefore, the material heterogeneity emphasizes the importance of studying the relationship between the cross-section geometry, manufacturing technology, load-bearing capacity, and failure mechanisms of pultruded FRP materials in engineering applications [19].

The mechanical behavior investigations of pultruded GFRP axially loaded columns are often [10,20,21,22,23,24,25,26], but there are only a few reports on the mechanical properties of the end supports. Combining pultruded profiles with the concrete core can diminish the GFRP heterogeneity problem [27,28]. However, the support problem of GFRP columns might become crucial when the profile serves as a load-bearing component. Additionally, the material’s failure mechanisms significantly influence the extent of reduction in the load-bearing capacity of the profile. For instance, the experimental results from references [29,30] indicate the phenomenon of end crushing and bugle fracture in GFRP profiles when used as concrete jackets, particularly pronounced in short columns under compression. Still, the end-support behavior isolates the fracture mechanisms of GFRP material. Most experimental studies on the axial compressive behavior of GFRP profiles [9,30,31,32,33,34] employed specific methods to constrain or reinforce the column ends, such as internal concrete filling or using steel fixtures at the ends. It is well known that direct loading of GFRP profiles can lead to premature failure at the end faces. However, few reports study the effects of adding or removing constraints at the end support locations. Only reference [34] inspects the various end constraints in GFRP circular column axial compression tests.

In addition, the mechanical performance of GFRP profiles from different manufacturers is different [19,35,36]. However, only a few studies have investigated the manufacturing defects of pultruded GFRP profiles. For example, Poulton and Sebastian [19] investigated mat misalignments in pultruded GFRP bridge decks. References [37,38] utilized Gaussian statistical models to analyze and locate manufacturing defects in pultruded GFRP profiles. These studies also examined the impact of manufacturing defects on these profiles’ failure mechanisms and mechanical performance reliability. This provides a valuable approach for studying the initial defects of GFRP profiles. Still, references [37,38] focused on relatively long elements, not considering the localized material failure in the support regions. Thus, the local GFRP resistance must be revised to investigate the local failure mechanisms essential for developing reliable manufacturing technologies, design methods, and numerical models [39,40].

At the same time, FE simulations, typical for analyzing the axial compressive behavior of GFRP profiles [18], employ two-dimensional (2D) shell homogeneous elements [31,37,38]. These finite elements can effectively represent the buckling and deformation states of GFRP columns. However, they cannot adequately capture the mechanical behavior of the support joints [18]. The three-dimensional (3D) shell elements used in reference [32] provide a more precise and detailed description of the failure of the square tube walls. Compared to 2D, 3D elements offer a more intuitive and accurate description of damage in the local regions of small-sized components.

This paper conducts axial compression tests on pultruded GFRP square tubular profiles and analyzes the impact of heterogeneity on the mechanical behavior of pultruded profiles. To distinguish between the material properties of GFRP pultruded profiles and the mechanical characteristics of square tubular components, the authors performed experimental analysis on pultruded GFRP square tubes with identical cross-sections but different slenderness ratios (i.e., 6, 10, and 15). This test program verifies a simplified numerical model and determines the material failure mechanisms behind the mechanical performance of the pultruded profiles for various column lengths.

## 2. Test Program

### 2.1. Material and Methods

The pultruded GFRP square tube samples (40 mm width and 3 mm wall thickness) and material test specimens used in this study were manufactured by Henan Embrace Composite Materials Co., Ltd. (Henan, China). The material tests followed the general principles of the standards GB/T 1446-2005 [41], GB/T 1447-2005 [42], and GB/T 1448-2005 [43]. Material tests were conducted using the manufacturer-provided standard pultruded GFRP specimens. Each group of specimens consisted of five samples. The Byes2100 universal testing machine (Bangyi Precision Measuring Instruments, Shanghai, China) was used for loading, with a uT7116Y static high-speed strain gauge (Utekl Electronic Technology, Wuhan, Hebei, China) to record the strain gauge date. Displacement control was employed for loading at a 2 mm/min rate. Table 1 summarizes the material characterization results.

### 2.2. Description of the Column Samples

The tubular specimens are grouped into three categories based on the slenderness ratio *λ*, namely S-6 (*λ* = 6), S-10 (*λ* = 10), and S-15 (*λ* = 15). These columns have nominal lengths of 90 mm, 150 mm, and 230 mm. The compressive tests consist of two stages. The first round considers the columns without the additional support of the loaded ends and employs a single column for each category. The unidirectional orientation of glass filaments and the lack of transverse confinement cause a massive failure localized at the column support sections. Therefore, the first testing stage provides a more intuitive exploration of the heterogeneity effect on the mechanical behavior of pultruded GFRP profiles. The second stage employs the same loading conditions. However, the end constraints (Figure 1) protect the support joints, following the approach described in the reference [10]. Figure 1 shows the supporting plate’s geometry parameters. Five columns of each *λ* category belong to the second testing stage.

### 2.3. Test Method

The experiments use a YAW-600C computer-controlled servo-hydraulic universal testing machine (Tengjie Instrument Equipment, Jinan, China), as shown in Figure 2a. The loading method employs displacement control with a 1 mm/min rate. A sudden deformation increase terminates the column tests. Figure 2b,c shows the typical compression test results. This study collects load and deformation records, testing 18 column samples. Strain gauges placed on all four sides of the middle sections of the specimen walls measure deformations; a uT7116Y device collects the data every second. The loading machine also records load and deformation data.

## 3. Test Results

### 3.1. The First Test Stage: Control Samples

Figure 3 shows the compression test results of the control specimens. In the absence of end constraints, the primary failure mode of the square tube profile involves crushing the matrix with the fibers scattering outward in a dispersed fashion at the end support. The material crushing at the support continues with the load, slowly progressing from the support toward the column body. At the same time, the specimen’s body shows no apparent damage. These results are consistent with the test observations [10]. In other words, the stress distribution in the GFRP columns concentrates at the supports without end constraints, causing premature material failure. The crushing failure of the support end leads to delamination between the GFRP fibers and the matrix, so the actual failure mode of the GFRP material is quite complex. Figure 3b,c also demonstrates delamination signs when cracks in the wall stimulate the fibers and matrix fragments to propagate on both sides along the thickness direction. Still, the profiles retain a residual load-bearing capacity even after the end material is crushed, exhibiting a pseudo-“yielding” stage similar to metal materials. This feature holds particular value in engineering applications.

### 3.2. The Second Test Stage: Samples with End Supports

Figure 4 shows the typical failure results of the profiles with the 6, 10, and 15 slenderness ratios. The photos depict the separation tendency of the flat components of the columns, with local and pronounced buckling in the walls. These results also proclaim the brittle nature of the pultruded GFRP material. In addition, all specimens exhibited internal matrix fracture sounds in the initial loading stage. Compared to homogeneous steel, the heterogeneous nature of GFRP makes its failure stochastic and hardly predictable.

Figure 5 shows the test results of all column samples with end supports. The primary failure mechanism is localized at the junction of the tube walls. This observation proves the profile separation tendency into flat parts. However, the damage localization process depends on the column length (or slenderness ratio). Thus, the failure of the S-15 samples is concentrated around a third of the column. The S-6 samples exhibit larger failure zones, with some specimens experiencing complete through-thickness failure (S-6-5). In addition, the failure mechanisms of specimens S-6-3 and S-6-4 involve crushing the material at the supports, which indicates a typical strength limit. Still, the failure does not exhibit apparent spreading as support plates constrain the end positions. The S-10 specimen group shows failure zones occupying approximately half the column height.

## 4. Discussion of the Test Results

Adding the end constraints changes the failure mechanisms of the GFRP profiles and moves the failure zones from the end section to the column body (Figure 5). Table 2 describes the ultimate resistance of the column samples. In this table, the second column determines the loading results of the control samples (Section 3.1); the remaining columns correspond to the outcomes of the columns with supported ends (Section 3.2). Compared to specimens without the end constraints (Figure 3), the load-bearing capacity of the columns is noticeably enhanced. However, the failure of all specimens with the end supports occurred instantaneously. Thus, adding end constraints enhances the load-bearing capacity but increases the brittleness of the structural element.

The load-bearing capacity of the control samples (Table 2) does not follow a consistent pattern with column length. This is due to the heterogeneity of pultruded GFRP materials, which causes stress concentration at the end section during compression. In contrast, the distribution of fibers and matrix within the section is random. Positions with relatively lower fiber content within the section tend to fail first, ultimately exacerbating the randomness of the load-bearing capacity distribution. The test results of the columns equipped with support plates indicate a negative correlation between the load-bearing capacity and the column length. Thus, the S-10 and S-15 specimen groups did not fully exploit the mechanical properties of GFRP material. In contrast, the S-6 specimens reached the strength limit and demonstrated the efficient utilization of the GFRP material expressed in terms of the theoretical resistance of the reinforced composite [40].

Figure 6 shows the load-displacement diagrams of all columns. The red curves correspond to the test results of the columns with end supports, while the black curves represent the control outcomes. Figure 6 demonstrates the brittleness of GFRP material with a rapid loss of residual resistance after reaching the maximum load. In contrast, the control specimens exhibit a post-failure residual load-bearing plateau because of the local crushing of the support material, while the column body remains undamaged. In Figure 6a,b, the black curves show significant displacement, resembling a “yielding” stage typical for metal materials. In Figure 6c, the black curve depicts fluctuations in the residual resistance with increasing deformations. Due to the uneven distribution of damage across the support section, certain regions undergo failure preferentially, resulting in a slightly eccentrically compressed state. As the end damage progresses, the damaged section returns to a more “uniform” stress distribution state, leading to the observed fluctuating trend on the graph. These observations confirm the influence of material heterogeneity on the failure mechanisms of GFRP and support the previous findings [36,44,45].

The considerable deformation after reaching the ultimate resistance of the GFRP column enhances its safety in engineering applications. However, the noticeable increase in compressive resistance due to end constraints also intensifies the brittleness of the GFRP column, reducing the range of “allowable damage deformation.” Therefore, further research is valuable in exploring methods to increase the permissible deformation after the ultimate load while fully exploiting the load-bearing capacity of GFRP columns.

The previous results [35,40] demonstrated the decisive contribution of numerical methods to evaluating the mechanical efficiency of composite systems. Therefore, this study involves a finite element (FE) analysis tool to reveal the mechanisms behind the mechanical resistance of the support joints of GFRP columns.

## 5. Numerical Analysis of the Compressive Load Resistance Mechanisms

### 5.1. Description of the Simplified FE Model

This study employs the FE software ABAQUS v16.11.16 to analyze the mechanisms behind the mechanical resistance of the GFRP columns. The analysis considers three characteristic cases: column S-6 with and without the support plates and element S-15 with the supported ends. The model utilizes anisotropic elastic material parameters, which correspond to the values in Table 1. In other words, the material model does not adopt any failure criteria. Table 3 describes the model parameters. In this table, *E* determines the modulus of elasticity; *G* is the shear modulus; *η* is the Poisson’s ratio; the subscripts define the orthonormal directions with “1” oriented along the pultrusion pathway. The ABAQUS/Explicit solver is employed to solve the 3D deformation problem; the C3D8R solid elements are used for this purpose. Figure 7 shows the FE mesh of the column samples and the steel support with the average 4 mm size of the finite elements. Figure 8 shows the boundary conditions and assembly of the column samples. The loading conditions employ the deformation control, corresponding to the testing layout (Section 2.3).

The “hard contact” condition with a 0.2 friction coefficient simulates the contact between the constraints and the profile (Figure 8b,c). The coupling points of the profile and the support plate centroids are used to prevent separation between the constraints and capture the load and displacement data of the column. The column length is the only difference between the models in Figure 8b,c.

### 5.2. Simulation Results

Figure 9 shows the load-displacement diagrams of the selected column samples. These results demonstrate that Dynamic, Explicit analysis can more or less adequately simulate the deformation response of the pultruded GFRP columns. Table 4 summarizes the ultimate resistance prediction results, where Δ is the relative difference between the numerical and experimental outcomes. The following modeling aspects need consideration in this regard:**A difference in “elastic” deformations** results from the absence of the preloading stage in the physical tests. This difference increases when introducing the end plates (compare Figure 9a,b). This preloading is necessary to tighten the assembly parts; it is mandatory for complex loading setups, e.g., additional support plates.**The end plate effect.** The model adequately reflects deformations of the control sample (Figure 9a); the ultimate load prediction error is 7% (Table 4). However, the model cannot capture the support plate contribution to improving the mechanical resistance of the column samples, as evident in physical tests (Figure 9b). This discrepancy results from the essential effect of the plate on improving the material performance of the end zones, which is beyond the limited ability of the simplified numerical model.**The column length effect.** Table 4 indicates that the simplified model (neglecting the material heterogeneity) fails to predict the load-bearing capacity of the S-6 specimen with end supports, with a relative error of 18%. However, increasing the column length remedies the prediction results and reduces the error to 7%. These results relate the model nonlinearity to the buckling effects predominant in the S-15 sample.

The above considerations require analyzing the predicted deformation shapes of the columns. Figure 10 shows the corresponding simulation results, including stress distribution in the profiles. Notwithstanding the adequate deformation diagram in Figure 9a, the predicted deformation shapes of the control sample (Figure 10a) do not conform to the test result (Figure 4a). The apparent difference results from the local failure of the GFRP material at the supports, whose prediction is beyond the simplified model’s ability.

Figure 10b demonstrates more or less adequate stress distribution with the stress concertation at the profile edges. Still, the model fails to represent the experimentally observed tendency of profile separation on the flat plates (Figure 4b). In addition, the multiple buckling waves formed in the model (Figure 10b top) reduce the ultimate resistance, which does not reach the experimental outcomes (Figure 9b).

Figure 9c and Figure 10c reveal the model’s ability to predict both ultimate capacity and deformation shapes of the S-15 sample because of the reduction of material nonlinearity effects with the column length increase. For instance, the deformed shape in Figure 10c (bottom view) agrees well with the test observation in Figure 4c.

Thus, the simplified simulation model fails to capture the heterogeneous properties of the GFRP material. The test results show that the short GFRP columns do not exhibit pronounced buckling deformations. Thus, the simplified material model, lacking consideration for material failure criteria, cannot simulate the mechanical behavior of support joints of pultruded GFRP columns. Therefore, developing adequate material models determines the further research objective.

### 5.3. Physically Non-Linear Modeling

To illustrate the inconsistency of the above-discussed simplified modeling approach, this subsection presents the FE modeling results assuming the physically non-linear material model. This model employs the 3D Hashin [46] and Puck [47] failure criteria, describing fibers and polymer matrix damage processes. Table 1 determines the physical parameters of the model. These simulations are only demonstrative; the reference [45] provides detailed model descriptions. All remaining modeling parameters and setups stay the same, as described in Section 5.1.

Figure 11 shows the results of the S-6 control and alternative column samples, similar to Figure 10a,b. With the incorporation of material failure criteria, the stress distribution in Figure 11 is more uniform compared to Figure 10. In addition, the updated model predicts a more realistic deformation shape of the S-6 column (Figure 11b). This model intuitively depicts the failure behavior of the S-6 column, closely approximating the cracking patterns in Figure 5a. The bulging expansion in the midsection of the tube wall leads to a loss of lateral constraint at the wall edges.

By comparing Figure 11a,b, it is evident that the addition of end supports shifted the region of brittle failure in the GFRP, which reflects the compression test results. The failure initiation, originally occurring at the bottom of the short column, propagated from the midsection. This observation indicates that the end supports effectively protect the support joint of the GFRP column from brittle failure.

The red dashed lines in Figure 9a,b show the load-deformation diagrams of the FE model after incorporating the material failure criteria. The model demonstrates an almost linear deformation response until the ultimate resistance is reached. This outcome fits the expected elastic deformation mechanisms of the GFRP columns [18]. The non-elastic deformations of the column samples during the physical tests resulted from the absence of the preloading stage, which is necessary to tighten the assembly parts [48,49,50]. When the stresses in the model undergo the strength limit, cracking of the GFRP material causes the load-bearing capacity loss (Figure 11), aligning with experimental observations in Figure 9. Still, Figure 9b shows some residual stress in the model, which differs from the test result. This discrepancy suggests that introducing failure criteria alone into the simplified model only provides an intuitive reflection of the failure deformation pattern of the GFRP column, and additional measures are required to enhance the model’s adequacy.

The failure mechanism in the S-6 control sample model (Figure 11a) does not reflect the local crushing of the column end observed in the test (Figure 3a,b). This limitation indicates that introducing material failure criteria cannot remedy the numerical model adequacy because the tube wall consists of single-layer elements (Figure 7a), and the GFRP heterogeneity is neglected. A mesh refinement and the introduction of cohesive elements between meshes are necessary to simulate the internal failure mode of the tube wall [18,47].

### 5.4. Further Research

The above oversimplified numerical examples illustrate the limitations of the simplified model typical for structural design, e.g., [4,18]. In this context, the adequate prediction of the load-bearing capacity of the relatively long columns (Figure 9c) is remarkable. Reducing the column length increases the combined effect of the material heterogeneity and the support plates, which the simplified model failed to capture. The reference sample (“S6 control”) made this contradiction even more transparent. For instance, Figure 10a,b demonstrates no difference in the deformation patterns. This result is acceptable for homogeneous materials like steel or plain plastics, but Figure 3a and Figure 4a show entirely different failure mechanisms. Thus, the simple tests made the advanced material model necessary, even if the simplified model could provide a seemingly reasonable result (as the load-bearing capacity in the considered case). Introducing the GFRP failure criterion, mesh refinement, and cohesive finite elements may remedy the model’s adequacy [18,47]. Thus, further investigation should develop a model suitable for predicting GFRP failure, which is crucial for stress localization in structural connections.

Notwithstanding the simplicity of the test setup and loading layouts, the developed testing procedure can generate results for verifying physically non-linear numerical models and estimating the corresponding physical parameters of GFRP profiles. Still, preloading is mandatory to tighten the assembly parts and avoid non-elastic deformations observed in the tests until the ultimate load is reached (Figure 9). This condition is mandatory for further tests.

## 6. Conclusions

Pultruded GFRP material exhibits lower compressive capacity than steel structures because of the fibers’ inability to resist the compression stresses. The material’s heterogeneity also causes stress to concentrate mainly at the column ends under compression, making it prone to crushing failure under compression.End constraints substantially improve the load-bearing capacity of the tubular GFRP columns. This study provides a possible solution to improve the local behavior of the support joints. However, this complex setup requires preloading to tighten the assembly parts.The primary failure mode of pultruded GFRP tubular profiles is a failure at the junction of the tube walls. Due to the material’s limited allowable deformation, the transverse tensile stress at the tube wall connection exceeds the strength limit. This leads to brittle failure and the profile separation forming flat components. This failure is unfavorable in engineering applications. Therefore, it is advisable to consider reinforcement solutions to enhance the transverse constraints of the locally produced pultruded materials.The test results of the columns equipped with support plates indicate a negative correlation between the load-bearing capacity and the column length. The short columns with a slenderness ratio of 6 fully realize the material strength of the GFRP profile. Increasing the column length reduces the column efficiency expressed in terms of the theoretical resistance of the reinforced composite. Therefore, GFRP columns require additional means to avoid premature stability loss.The simplified numerical model adequately predicts the ultimate resistance of relatively long columns (the average prediction error does not exceed 7%). However, the model fails to predict the effects beyond the elastic material model, e.g., failure characteristic of support joints of pultruded GFRP profiles. Therefore, developing adequate material models determines the further research object, and the proposed testing procedure helps generate the experimental data and verify the model’s adequacy.

## Figures and Tables

**Figure 1 materials-17-00153-f001:**
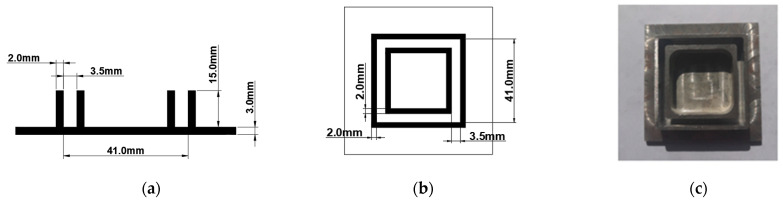
The geometry of the supporting plate for the column specimen: (**a**,**b**) the plate schematic; (**c**) the experimental view.

**Figure 2 materials-17-00153-f002:**
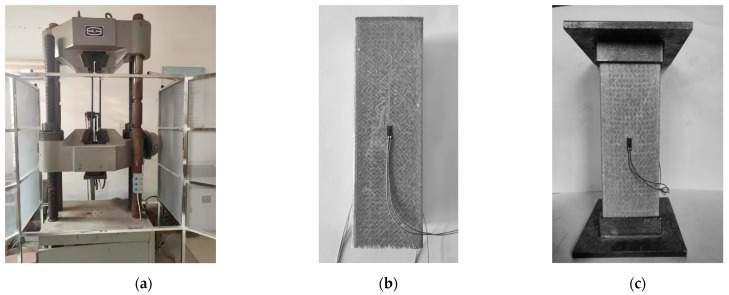
Loading device and specimens for the axial compression test of pultruded GFRP square tube profiles: (**a**) Loading device; (**b**) control specimen (λ = 6); (**c**) end supported specimen (λ = 6).

**Figure 3 materials-17-00153-f003:**
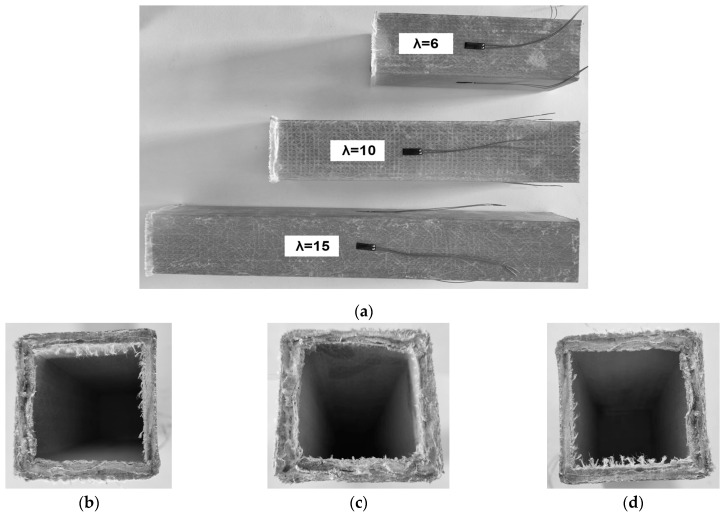
Test results of the control specimens: (**a**) Front view of the samples; (**b**–**d**) the support sections of S-6, S-10, and S-15 columns.

**Figure 4 materials-17-00153-f004:**
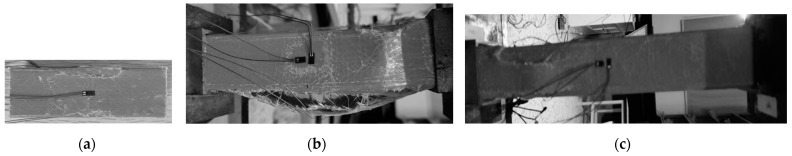
Characteristic failure of the square tubular columns: (**a**) S-6 series; (**b**) S-10; (**c**) S-15.

**Figure 5 materials-17-00153-f005:**
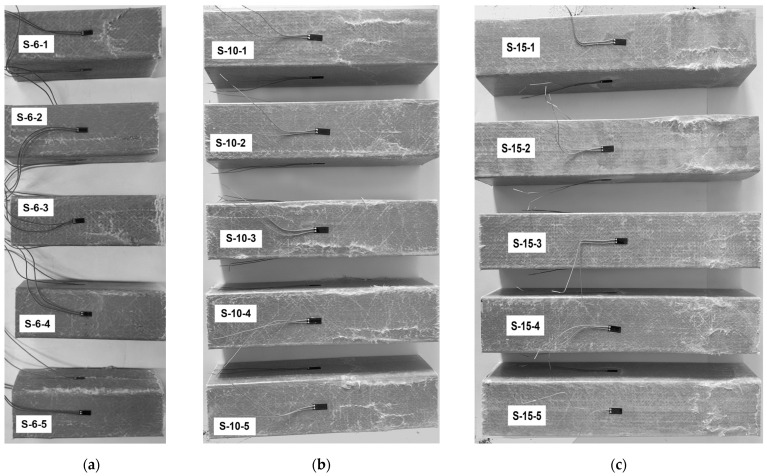
Test results of the columns with the end supports: (**a**) S-6 series; (**b**) S-10; (**c**) S-15.

**Figure 6 materials-17-00153-f006:**
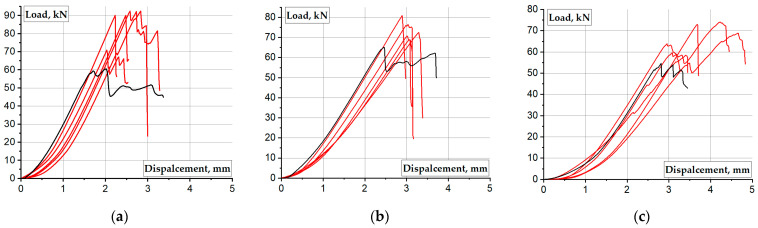
The load-displacement diagrams of GFRP columns: (**a**) λ = 6; (**b**) λ = 10; (**c**) λ = 15. Note: Black curve means control specimen without end constraints.

**Figure 7 materials-17-00153-f007:**
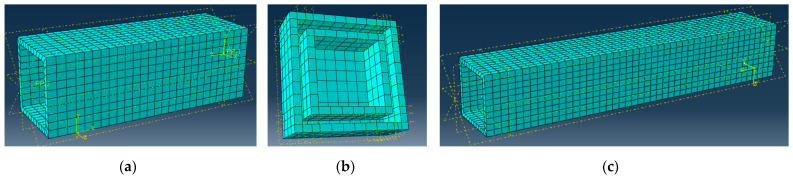
The mesh of the FE model: (**a**) S-6 sample; (**b**) support plate; (**c**) S-15 column.

**Figure 8 materials-17-00153-f008:**
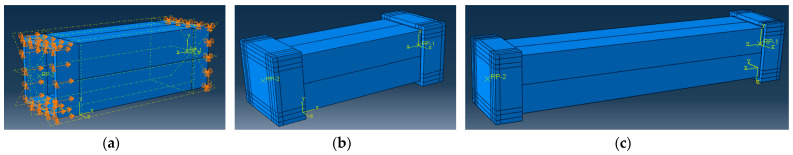
The simulation objects: (**a**) Boundary condition of the control S-6 sample; (**b**) the assembly model of the S-6 column; (**c**) the assembly model of the S-15 column.

**Figure 9 materials-17-00153-f009:**
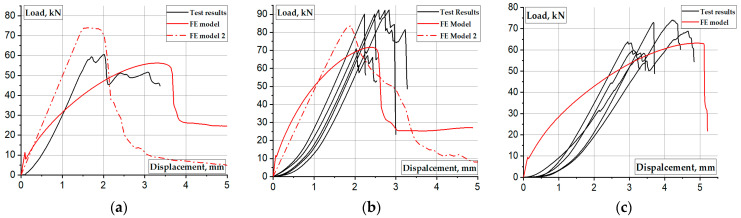
Simulated and experimental load-displacement diagrams: (**a**) S-6 control specimen; (**b**) S-6 sample; (**c**) S-15 column.

**Figure 10 materials-17-00153-f010:**
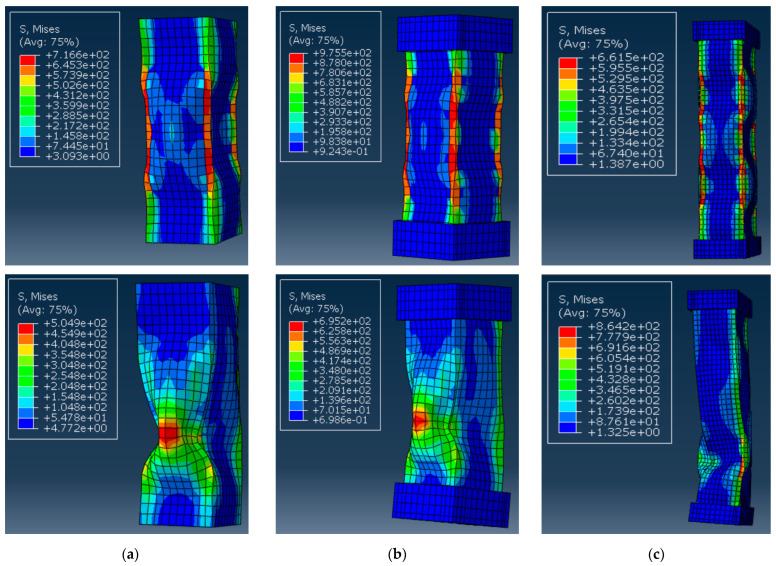
Simulated deformation shapes of the columns (ultimate load in the **top** view and maximum deformation in the **bottom** view): (**a**) S-6 control specimen; (**b**) S-6 sample; (**c**) S-15 column. Note: The stress measurement unit is MPa; the load and deformation values correspond to Figure 9.

**Figure 11 materials-17-00153-f011:**
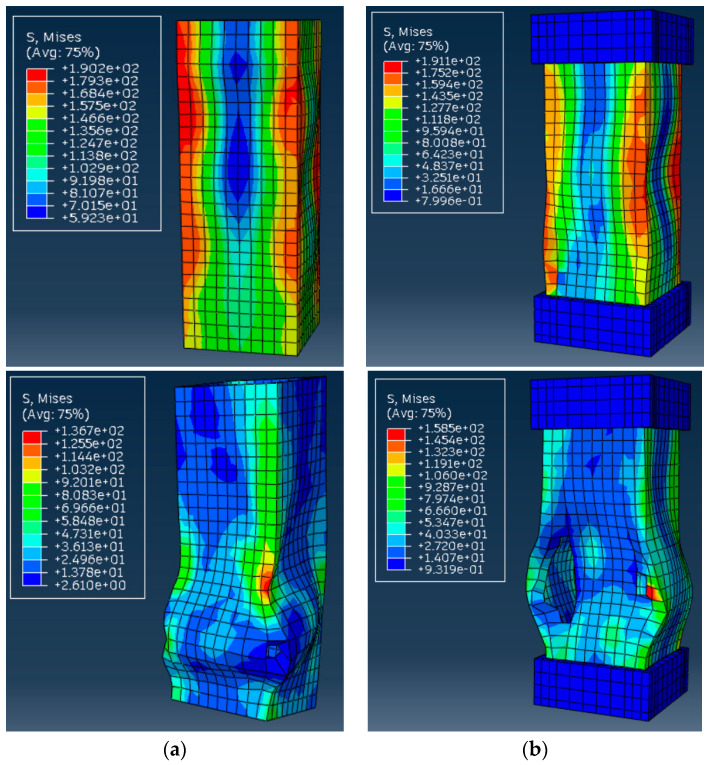
Simulated deformation shapes of the columns using physically non-linear material model (ultimate load in the **top** view and maximum deformation in the **bottom** view): (**a**) S-6 control specimen; (**b**) S-6 sample. Note: the stress measurement unit is MPa; the load and deformation values correspond to Figure 9 (“FE model 2”).

**Table 1 materials-17-00153-t001:** Mechanical properties of the GFRP material.

Parameter	Value *
Tensile strength (MPa)	464.2 ± 8.12%
Compressive strength (MPa)	189.2 ± 5.61%
Modulus of elasticity of longitudinal tension (GPa)	32.77 ± 1.54%
Modulus of elasticity of longitudinal compression (GPa)	53.30 ± 4.12%
Modulus of elasticity of transverse tension (GPa)	3.68 ± 6.85%
Modulus of elasticity of transverse compression (GPa)	6.16 ± 5.91%
Modulus of shear (GPa)	1.60 ± 8.90%
Poisson’s ratio (-)	0.27

* A number next to the symbol “±” represents the coefficient of variation for the five samples.

**Table 2 materials-17-00153-t002:** Failure load in pultruded GFRP square tube axial compression test (kN).

Type	P_*u*,*control*_	P_*u*,1_	P_*u*,2_	P_*u*,3_	P_*u*,4_	P_*u*,5_	Mean *
S-6	60.50	92.43	70.91	90.1	90.07	92.48	87.20 ± 10.5%
S-10	65.01	68.82	80.95	76.48	72.49	70.74	73.90 ± 6.6%
S-15	54.63	58.39	72.92	74.03	63.79	68.81	67.59 ± 9.7%

* The average value does not include the results of the reference columns.

**Table 3 materials-17-00153-t003:** The parameters of the GFRP material’s model.

*E*_1_ (GPa)	*E*_2_ (GPa)	*E*_3_ (GPa)	*G*_12_ (GPa)	*G*_13_ (GPa)	*G*_23_ (GPa)	*η* _12_	*η* _13_	*η* _23_
33.27	3.68	3.68	1.6	1.6	1.6	0.27	0.27	0.4

**Table 4 materials-17-00153-t004:** Verification of the ultimate load prediction results using the test data.

Column	Test Result * (kN)	Prediction (kN)	Δ (%)
S-6 control	60.5	56.3	6.9
S-6	87.2 ± 10.5%	71.8	17.7
S-15	67.6 ± 9.7%	63.2	6.5

* A number next to the symbol “±” represents the coefficient of variation for the five samples.

## Data Availability

Data are contained within the article.

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
