# Peer review of "The Material Heterogeneity Effect on the Local Resistance of Pultruded GFRP Columns"

_materials, 2023, doi:10.3390/ma17010153_

Round 1
Reviewer 1 Report
Comments and Suggestions for Authors
The paper
“The Material Heterogeneity Effect on the Local Resistance of Pultruded GFRP Columns”,
By Yongcheng Zhu et al.,
examines the failure performance of locally produced pultruded GFRP (glass fiber-reinforced polymer) materials, particularly focusing on their mechanical resistance in supporting joints of tubular GFRP columns.
In this sense, the concept is well-conceived, very interesting and in line with the scope of the Journal. It also has real-life implications.
Nevertheless, some conceptual and editorial concerns should be addressed before granting full acceptance. These are enlisted here below.
1. Arguably, the main limitation of the proposed research work is that the proposed simplified FE model overlooks the nonlinearity of GFRP material. This is a strong limiting assumption, that however can be made for specific cases and applications. However, it must be thoroughly justified.
2. Related to the first main remark, relatively short rectangular profile fragments were used for the experimental part. This particular choice could be very limiting since e.g. does not allow the investigation of the nonlinear geometric effects induced by the GFRP well well-known high deformability under static and dynamic loads.
3. Again related to the first remark, the results of the simulations, shown in Figure 9, do not match well the experimental behaviour. Specifically, the initial (pre-peak) behaviour of the GFRP clearly shows a hardening phase, while the simulation clearly undergoes softening. This different trend should be discussed and explained and/or motivated.
4. The introduction section lacks a proper state-of-the-art review. The scientific literature concerning the structural performance and damage resilience of pultruded GFRP beams and columns is quite extensive, with recent examples such as the works Treed Gaussian process for manufacturing imperfection identification of pultruded GFRP thin-walled profile, Recursive partitioning and Gaussian Process Regression for the detection and localization of damages in pultruded Glass Fiber Reinforced Polymer material, and related ones. These could be included in the text of the paper.
5. Related to the above remark, with only 11 pages, the paper is quite short. Without being uselessly lengthy and wordy, it will be useful to add more details about the context of this research, the execution of the tests, and the discussion of the results.
6. Especially the Conclusions should be better introduced; limiting the discussion to a bullet point list is quite too concise.
Comments on the Quality of English LanguageThe English is overall good
Author Response
Comment 0: The paper <…> examines the failure performance of locally produced pultruded GFRP (glass fiber-reinforced polymer) materials, mainly focusing on their mechanical resistance in supporting joints of tubular GFRP columns. In this sense, the concept is well-conceived, very interesting, and in line with the scope of the Journal. It also has real-life implications.
Nevertheless, some conceptual and editorial concerns should be addressed before granting full acceptance. These are enlisted here below.
Answer: The Authors express their sincere gratitude to the Reviewer for the positive evaluation of the submission. They also sincerely thank the Reviewer for sharing time and knowledge. The manuscript was improved by implementing all suggestions by the Reviewer – the yellow color highlights all modifications in the text.
Comment 1: Arguably, the main limitation of the proposed research work is that the proposed simplified FE model overlooks the nonlinearity of GFRP material. This is a strong limiting assumption that, however, can be made for specific cases and applications. However, it must be thoroughly justified.
Reply: The Authors intended to illustrate the limitations of the simplified model typical for structural design, e.g., [https://doi.org/10.1038/s41598-022-20666-x], and determine its applicability range. In this context, the adequate prediction of the load-bearing capacity of the relatively long columns (Figure 9c) is remarkable. Reducing the column length increases the combined effect of the material heterogeneity and the support plates, which the simplified model failed to capture. The reference sample (“S6 control”) made this contradiction even more transparent. For instance, Figures 10a and 10b demonstrate no difference in the deformation patterns. This result is acceptable for homogeneous materials like steel or plain plastics, but Figures 3a and 4a show entirely different failure mechanisms. Thus, the simple tests made the advanced material model necessary, even if the simplified model could provide a seemingly reasonable result (as the load-bearing capacity in the considered case).
At the same time, the Authors appreciate and accept the pointed presentation shortcoming. They extended Section 5 by introducing the physically nonlinear modeling example of the S-6 samples (control and alternative), which makes the discussion more objective and illustrative.
Correction in the manuscript:
- Section 5.3 was introduced to present the physically nonlinear modeling example of the control sample S-6.
- The updated Figures 9a and 9b include the physically nonlinear modeling result.
- New Figure 11 and the corresponding discussion in Section 5.3 provide the objective rationale of the need for the advanced modeling concept in the fifth conclusion.
- Section 5.4 determines the prospective research.
Comment 2: Related to the first main remark, relatively short rectangular profile fragments were used for the experimental part. This particular choice could be significantly limiting since, e.g., it does not allow the investigation of the nonlinear geometric effects induced by the GFRP well-known high deformability under static and dynamic loads.
Reply: The eminent Reviewer is generally correct, and this comment aligns with the above considerations. Still, the material nonlinearity neglection in the geometrically nonlinear model (used in this study) does not reduce the prediction adequacy of the S-15 column samples in this experimental program. The length reduction ensures turning the problem into a material issue, revealing the importance of the FRP heterogeneity. In particular, this issue is relevant to FRP joints (localizing stresses). Still, the typical tests, e.g., [https://doi.org/10.3390/ma15082901], cannot allow isolating the FRP fracture and making it a simulation object straightforward. The short columns (the shortest samples in this study) solved this problem. In addition, the support plates make the testing limitations apparent besides the simulation problem, which is unsolvable, neglecting the material nonlinearity. Therefore, the authors extended the simulation example by introducing the material model from the previous study [https://doi.org/10.3390/ma15082901] and made the discussion more objective.
Comment 3: Again, related to the first remark, the results of the simulations, shown in Figure 9, do not match the experimental behavior well. Specifically, the initial (pre-peak) behavior of the GFRP clearly shows a hardening phase while the simulation undergoes softening. This different trend should be discussed and explained and/or motivated.
Reply: The Authors believe the answers above have clarified the modeling issue. On the other hand, the explanation is straightforward regarding the disagreement between the modeling and test results in the pre-peak stage. The non-elastic deformations of the column samples during the physical tests resulted from the absence of the preloading stage, which is necessary to tighten the assembly parts, e.g., [https://doi.org/10.1002/pc.27237, https://doi.org/10.1177/09544054231188997].
Correction in the manuscript: The corresponding discussion was added in Sections 5.3 and 5.4.
Comment 4: The introduction section lacks a proper state-of-the-art review. The scientific literature concerning the structural performance and damage resilience of pultruded GFRP beams and columns is quite extensive, with recent examples such as the works Treed Gaussian process for manufacturing imperfection identification of pultruded GFRP thin-walled profile, Recursive partitioning and Gaussian Process Regression for the detection and localization of damages in pultruded Glass Fiber Reinforced Polymer material, and related ones. These could be included in the text of the paper.
Reply: The Authors sincerely appreciate this comment and the suggested object for discussion.
Correction in the manuscript:
- The Authors substantially extended the literature review (Lines 66–99), discussing the structural behavior of relatively long and short columns, fabrication processes, and numerical modeling aspects.
- Twelve new references (i.e., [18,19,27-34,37,38]) were added in the literature review.
Comment 5: Related to the above remark, with only 11 pages, the paper is quite short. Without being uselessly lengthy and wordy, it will be useful to add more details about the context of this research, the execution of the tests, and the discussion of the results.
Reply: The Authors accepted this comment. They extended the literature analysis and added the numerical simulation example (Section 5.3); Section 5.4 discusses the model limitations and describes the research prospects.
Correction in the manuscript:
- The above modification prolonged the text by 1153 words (3.5 pages).
- A new section (“4. Further research”) was introduced.
Comment 6: The Conclusions should be introduced better; limiting the discussion to a bullet point list is too concise.
Reply: The Authors appreciate this note. However, they prefer the bulleted statements form, preserved in the updated manuscript. Still, Section 5.4 discusses the model limitations and describes the research prospects. The Authors hope that the above corrections made the manuscript acceptable for publication.
Reviewer 2 Report
Comments and Suggestions for Authors
1.Please add more citations on the introduction part.
2. How does the incorporation of end constraints impact the failure mechanisms and load-bearing capacities of the GFRP columns compared to the control samples, and what implications does this shift hold for the structural behavior of the material?
3.Could you elaborate on the observed discrepancies in load-bearing capacities concerning column length in the tested GFRP columns, particularly with respect to stress concentration, fiber distribution, and the influence of support plates?
Author Response
Comment 1: Please add more citations to the introduction part.
Reply: The Authors appreciate this suggestion.
Correction in the manuscript:
- The Authors substantially extended the literature review (Lines 66–99), discussing the structural behavior of relatively long and short columns, fabrication processes, and numerical modeling aspects.
- Twelve new references (i.e., [18,19,27-34,37,38]) were added in the literature review.
Comment 2: How does the incorporation of end constraints impact the failure mechanisms and load-bearing capacities of the GFRP columns compared to the control samples, and what implications does this shift hold for the structural behavior of the material?
Reply: Most experimental studies on the axial compressive behavior of GFRP profiles [https://doi.org/10.1016/j.tws.2018.01.031, https://doi.org/10.1016/j.conbuildmat.2021.125353, https://doi.org/10.1016/j.compstruct.2021.113650] employed specific methods to constrain or reinforce the column ends, such as internal concrete filling or using steel fixtures at the ends. It is well-known that direct loading of GFRP profiles can lead to premature failure at the end faces. However, few reports study the effects of adding or removing constraints at the end support locations. Only reference [https://doi.org/10.1016/j.tws.2015.06.019] inspects various end constraints used in GFRP circular column axial compression tests.
To illustrate the inconsistency of the simplified modeling approach used in the manuscript, the newly introduced section (Section 5.3) presents the FE modeling results assuming the physically non-linear material model. This model employs the 3D Hashin and Puck failure criteria, describing fibers and polymer matrix damage processes. Newly generated results of the S-6 control and alternative column samples similar to Figures 10a and 10b are shown in Figure 11. By comparing Figures 11a and 11b (which correspond to the control sample and the column with the support plates), it is evident that the addition of end supports shifted the region of brittle failure in the GFRP, which reflects the compression test results. The failure initiation, originally occurring at the bottom of the short column, propagated from the midsection. This observation indicates that the end supports effectively protect the support joint of the GFRP column from brittle failure.
This paper conducts axial compression tests on pultruded GFRP square tubular profiles and analyzes the impact of heterogeneity on the mechanical behavior of pultruded profiles. It aims to develop a testing procedure that helps generate the experimental data and verify a numerical model’s adequacy, focusing on the non-linear material failure mechanisms.
Correction in the manuscript: The corresponding clarifications were added in the Introduction and Sections 5.3 and 5.4.
Comment 3: Could you elaborate on the observed discrepancies in load-bearing capacities concerning column length in the tested GFRP columns, particularly with respect to stress concentration, fiber distribution, and the influence of support plates?
Reply: GFRP material heterogeneity controls the column resistance. In this context, the adequate prediction of the load-bearing capacity of the relatively long columns (Figure 9c) is remarkable. Reducing the column length increases the combined effect of the material heterogeneity and the support plates, which the simplified model failed to capture. The reference sample (“S6 control”) made this contradiction even more transparent. For instance, Figures 10a and 10b demonstrate no difference in the deformation patterns. This result is acceptable for homogeneous materials like steel or plain plastics, but Figures 3a and 4a show entirely different failure mechanisms. Thus, the simple tests made the advanced material model necessary, even if the simplified model could provide a seemingly reasonable result (as the load-bearing capacity in the considered case). The Authors hope that the additional non-linear modeling example in Section 5.3 and the corresponding discussions clarify the GFRP resistance mechanisms and the support plates’ effect on the failure of short-column samples.
Acknowledgment. The authors sincerely appreciate the time and knowledge shared by this Reviewer. Comments and suggestions contributing to improving the manuscript are also genuinely appreciated. The yellow color highlights all corrections in the text.
Round 2
Reviewer 1 Report
Comments and Suggestions for Authors
This Reviewer is overall satisfied with the changes made by the Authors and the reply to the first round of comments, especially the first one, the addition of FE Model 2, and the addition of Subsection 5.3
Hence, this Reviewer has no further major comments, except to grammar check the whole manuscript during the proofreading phase
Comments on the Quality of English LanguageThe English of the manuscript has been improved since the first draft of the paper.